# Food Service Kitchen Scraps as a Source of Bioactive Phytochemicals: Disposal Survey, Optimized Extraction, Metabolomic Screening and Chemometric Evaluation

**DOI:** 10.3390/metabo13030386

**Published:** 2023-03-05

**Authors:** Tatiana de Souza Medina, Carolina Thomaz dos Santos D’Almeida, Talita Pimenta do Nascimento, Joel Pimentel de Abreu, Vanessa Rosse de Souza, Diego Calandrini Kalili, Anderson Junger Teodoro, Luiz Claudio Cameron, Maria Gabriela Koblitz, Mariana Simões Larraz Ferreira

**Affiliations:** 1Laboratory of Bioactives, Food and Nutrition Graduate Program, Federal University of State of Rio de Janeiro (UNIRIO), Rio de Janeiro 22290-250, Brazil; 2Center of Innovation in Mass Spectrometry, Laboratory of Protein Biochemistry, Federal University of State of Rio de Janeiro (UNIRIO), Rio de Janeiro 22290-250, Brazil; 3Laboratory of Functional Food, Food and Nutrition Graduate Program, Federal University of State of Rio de Janeiro (UNIRIO), Rio de Janeiro 22290-250, Brazil; 4Laboratory of Biotechnology, Food and Nutrition Graduate Program, Federal University of State of Rio de Janeiro (UNIRIO), Rio de Janeiro 22290-250, Brazil

**Keywords:** fruit and vegetable non-edible parts, waste, health-promoting foods, metabolomics, bioactive compounds

## Abstract

Untargeted metabolomics is a powerful tool with high resolution and the capability to characterize a wide range of bioactive natural products from fruit and vegetable by-products (FVB). Thus, this approach was applied in the study to evaluate the phenolic compounds (PC) by metabolomic screening in five FVB after optimizing their extraction. The total phenolic content and antioxidant activity analyses were able to select the best extractor (SM) and ultrasonication time (US) for each FVB; methanol was used as a control. Although ultrasonication yielded a lower number of PC identifications (84 PC), the US extract was the most efficient in total ionic abundance (+21% and +29% compared to the total PC and SM extracts, respectively). Ultrasonication also increased the phenolic acid (+38%) and flavonoid classes (+19%) extracted compared to SM, while the multivariate analyses showed the control as the most dissimilar sample. FVB extracted from the same parts of the vegetable/fruit showed similarities and papaya seed presented the most atypical profile. The application of the metabolomics approach increased the knowledge of the bioactive potential of the evaluated residues and possibilities of exploring and valorizing the generated extracts.

## 1. Introduction

Fruits and vegetables play a crucial role in the human diet, with the recommended daily consumption of at least 400 g per day [1]. In international trade, this group represents approximately 8% of global food production and they are among the most valuable agricultural commodities [2]. Concomitant with high production, the residual mass increases as well. However, the residual mass resulting from the non-edible structural parts of fruits and vegetables (stems, leaves, peels, pulp, seeds, and roots) still contain a large number of natural products with high bioactivity, including phenolic compounds [3,4,5,6,7,8].

Epidemiological studies have associated the usual intake of fruits and vegetables with health promotion and reduced mortality risks from chronic non-communicable diseases [9]. In fact, fruits and vegetables are sources of phytochemicals with biological properties such as protective effects against oxidative damage, inflammatory conditions, dyslipidemia, cancer, diabetes, and hypertension, among others [10,11].

Given the interest in bioactive phytochemicals in different fields, the non-edible fractions of fruits and vegetables could be processed and converted into extracts with applications as food additives, supplements, as well as pharmaceutical formulations [12]. The aforementioned alternatives could provide a way to reduce waste and generate promising high value-added products. Fruit and vegetable by-products (FVB) have been tested as sources of bioactive ingredients to replace synthetic antioxidants, such as butylated hydroxytoluene (BHT) and butylated hydroxyanisole (BHA), due to their natural appeal, low cost, and greater safety for human consumption [13]. Conventional methods and green technologies have been successfully used to extract different phenolic compounds from FVB, including ellagitannins, proanthocyanidins, and flavonoids from pomegranate peel [14,15,16]; anthocyanins from blueberry processing by-products [17]; agrimoniin from strawberry agro-industrial by-products [18]; hydroxycinnamic acid and anthocyanins from carrot root [19]; tannins, vanillin, and mangiferin in peel and seeds from mango [20]; flavonoids, saponins, tannins from lemon peel [21]; and polyphenols from onion skin [19,22]. Extracts from these residues have been incorporated into the production of functional foods, thereby increasing their nutritional value [23], and applied as food additives with antimicrobial, coloring, flavoring, and thickening properties [24].

However, future applications of by-products require studies on the conditions of extraction due to the complexity of vegetable matrices [25]. Natural products have distinct polarities and a wide variety of chemical properties associated with their structures, which are reflected by the difficulty in achieving optimized extractions for specific matrices [26]. Therefore, it is essential to understand how the extraction conditions influence the qualitative and quantitative profiles of phenolic compounds. Following this demand, studies involving factorial planning have been applied to optimize the best composition and proportion of solvents, time, temperature, and agitation necessary for the extraction of natural products with maximum bioactivity [27]. Furthermore, state-of-the-art extraction of phenolic compounds has also been enhanced by modern technologies. While conventional extraction techniques, such as Soxhlet, maceration, and hydro-distillation, have downsides such as low yields and the use of highly toxic organic solvents that may leave residues in the extract, green methods reduce energy consumption and apply environmentally beneficial organic solvents [28]. However, it is essential to study the factors that increase extraction yields and influence the profile of the phytochemicals of interest in FVB extracts [29].

Another important issue is to characterize the bioactive compounds and chemically measure their biological activity in vitro and/or in vivo [30]. With the advent of functional genomic approaches, metabolomics techniques have been widely applied in plant by-products [31,32]. This may provide as much information as possible on the profile and content of phenolic compounds in FVB. In this context, the aim of this work was to optimize the extraction of phenolic compounds from FVB and evaluate their antioxidant capacity in vitro and phenolic profile by UHPLC-ESI-QTOF-MS^E^ in order to valorize these residues.

## 2. Materials and Methods

### 2.1. Data Survey and Fruit and Vegetable by-Products (FVB) Selection

FVB were obtained from two different food services, one in a children’s shelter (Food Service A) and the other in a university refectory (Food Service B). The two locations were selected by convenience analysis (non-probabilistic sampling). The children’s shelter was located in the Northern region of Rio de Janeiro (Brazil) and served an average of 200 meals per day to pre-school and school-aged children plus staff. The university refectory was in the Southern region of the same city and served approximately 1000 meals per day (lunch and dinner) to young adults (undergraduate and graduate students) plus university staff. The two food services were evaluated for a period of 20 non-consecutive days. The volume of each sample was measured in a standardized plastic bag (total volume of 8.4 L) and then weighed on a digital scale. FVB were selected according to the criterion of higher mass or volume, but also by the characteristics of the sample regarding the extraction potential of the antioxidant/phenolic compounds through data available in the literature.

After collection, the residues were dried in a forced ventilation oven (Marconi, MA 035, Piracicaba, SP, Brazil) at 65 °C for 12 h, ground in a ball mill (SL-38, Solab, Paris, France) for 1 min, manually sieved (48 mesh, 300 µm), and weighed using an analytical scale. The final moisture content was determined in triplicate (AOAC, 1999) and the yield was calculated. The flours obtained from the samples were stored in plastic packaging in an ultra-freezer (−80 °C) until analysis.

### 2.2. Phenolic Compound Extraction

For the extraction of phenolic compounds, an experimental design was carried out according to Appendix A. First, a mixture simplex-centroid design was created using Statistica software (Statsoft version 7.0, Tulsa, OK, USA) to determine the ethanol and water ratio suited for extracting the highest concentration of phenolic compounds for each sample (0%, 25%, 50%, 75%, and 100% ethanol). The sample/solvent ratio was 100 mg mL^−1^. The solutions were then stirred (140 rpm, 30 min, at 25 °C) (Tecnal, Piracicaba, SP, Brazil), centrifuged (4 °C, 20 min, 10,000× *g*) (Thermo Scientific, Darmstadt, Germany), and the supernatants were collected. The total reducing capacity (TRC) determined using the Folin-Ciocalteu method and the antioxidant capacity determined using the DPPH method (both described in Section 2.3) were performed in triplicate for the selection of the best solvent, which was named “SM” (selected mixture).

After selecting the best solvent ratio, the application of ultrasonication as an auxiliary extraction method was evaluated (Appendix A). Probe ultrasonication (Desruptor 500 W, Eco-sonics, Indaiatuba, SP, Brazil) was applied for extraction three times: 5, 30, and 60 min (20 kHz, 375 W). Then, the extracts were centrifuged (4 °C, 20 min, 10,000× *g*) and the supernatants were collected. Again, the antioxidant capacity (AC) and TRC were used to select the best extraction time, which was named “US” (ultrasonication-assisted extract).

For comparison with the optimized extraction methods (SM + US), a sequential extraction (total phenolic content—TPC) consisting of the extraction of free (FPC) and bound phenolic compounds (BPC) was also carried out according to Santos et al. [33]. FPC were extracted from 70 mg of sample with 1 mL of 80% methanol and stirred at 25 °C (200 rpm, 10 min). After centrifugation (5000× *g*, 25 °C, 10 min), the supernatant was removed. Extraction was performed twice, and the obtained extracts were pooled together. The pellets resulting from extraction were submitted to alkaline and acid hydrolysis [33] to extract BPC. The sum of FPC and BPC was considered the total phenolic content (TPC) of each sample.

All extracts (SM, US, FPC, and BPC) were evaporated (SpeedVac Savant, Thermo Fisher Scientific, Waltham, MA, USA) and reconstituted in 1.5 mL of methanol, acetonitrile, and ultra-pure water (2:5:93, *v*/*v*/*v*), purified using the Milli Q-Millipore system (Millipore, Darmstadt, Germany). The reconstituted extracts were filtered (0.22 μm, hydrophilic PTFE) (Analitica, São Paulo, SP, Brazil) and stored in vials at −80 °C.

### 2.3. Determination of Antioxidant Capacity

The antioxidant capacity of samples was determined, in triplicate, using the DPPH (2,2-diphenyl-1-picrylhydrazyl) radical scavenging method, the ferric reducing antioxidant power method (FRAP), and the oxygen radical absorbance capacity method (ORAC), adapted for microplates [34]. For the DPPH method, a 20 μL aliquot of each extract was combined with 280 μL of the DPPH solution (32 μg/mL) and incubated (30 min, in the dark, 25 °C). The absorbance of the mixture was measured spectrophotometrically at 715 nm using a microplate reader (FlexStation III, Molecular Devices, San Jose, CA, USA). Methanol was used as a blank and the results were expressed as µmol Trolox (6-hydroxy-2,5,7,8-tetramethylchroman-2-carboxylic acid) equivalents (µmol TE g^−1^ of sample, dry basis). For the FRAP assay, the reagent was prepared with acetate buffer (0.3 M, pH 3.6), iron (III) chloride hexahydrate (FeCl_3.6_H_2_O, 20 mM), and TPTZ (2,3,5-triphenyl-tetrazolium chloride) solution (10 mM) in a 10:1:1 ratio. A 20 μL aliquot of each extract was combined with 15 μL of Milli-Q water and 265 μL of FRAP reagent, gently vortexed, and incubated (30 min, 37 °C). The absorbance of the mixture was measured spectrophotometrically at 595 nm using the FlexStation III microplate reader. Milli-Q water was used as a blank and the results were expressed as µmol FeSO_4_ (iron sulfate) equivalents (µmol FeSO_4_ g^−1^ of sample, dry basis). The ORAC analysis was performed by fluorimetry (Spectramax I3X, Molecular Devices, San Jose, CA, USA). Eighty µL of the fluorescein solution (80 nM) and 80 µL of the sample or blank (75 mM sodium phosphate buffer, pH 7.4) were added to the dark microplate. Then, 40 µL of 221 mM AAPH [2,2’-azobis(2-methyl-propanimidamide) dihydrochloride] was added. The antioxidant activity was monitored by a decrease in fluorescence measured at 485 nm (excitation) and 520 nm (emission) for 2 h. The antioxidant activity was determined by the area under the integrated fluorescence curve over time using GraphPad Prism software. The ORAC results were expressed as µmol Trolox (6-hydroxy-2,5,7,8-tetramethylchroman-2-carboxylic acid) equivalents (µmol TE g^−1^ of sample, dry basis).

The TRC was determined, in triplicate, according to Singleton et al. [35], adapted for microplates. An aliquot of extract (100 μL) was added to 700 μL of Milli-Q water in a test tube. After homogenization, 50 μL of Folin-Ciocalteu reagent and 150 μL of 20% sodium carbonate were added. The mixture was incubated (30 min, 40 °C) and 300 μL of the final solution was transferred to a microplate. The absorbance was measured at 750 nm using a FlexStation III microplate reader (Molecular Devices, San Jose, CA, USA). Solvent blank and standard curve analysis were performed with gallic acid (5 to 200 μg/L). The results were expressed in mg of gallic acid equivalents (EAG) per 100 g of sample, dry basis.

For the calculation of all spectrophotometric analyses described above (DPPH, FRAP, ORAC, and TRC), the absorbance of the blank was subtracted from the samples. Then, the resulting absorbance was plotted against the standard curve, and the concentration in each sample was calculated from the graph.

### 2.4. Metabolomics Analysis of FVB Phenolic Profile by UHPLC-MS^E^

The determination of the phenolic profile was performed by injecting 2 μL of each sample into an Ultra High Performance Liquid Chromatography (UHPLC) Acquity system (Waters, Milford, MA, USA) coupled with XEVO G2S Q-Tof (Waters, Wilmslow, UK) and equipped with ionization source electrospray. An UHPLC HSS T3 C18 column (100 × 2.1 mm, 1.8 μm particle diameter; Waters) was used at 30 °C with a flow rate of 0.5 mL/min of ultra-pure water containing 0.3% formic acid and 5 mM ammonium formate (mobile phase A) and acetonitrile containing 0.3% formic acid (mobile phase B) according to the following gradient: 0 min—97% A; 11.80 min—50% A; 12.38 min—15% A; 14.11—97% A. Data were acquired in triplicate in MS^E^ negative and centroid mode between *m/z* 50 and 1200; collision energy ramp from 30 to 55 V; cone voltage 30 V; capillary voltage 3.0 kV; desolvation gas (N_2_) 1200 L/h at 600 °C; cone gas 50 L/h; source at 150 °C; and using leucine enkephalin (Leu-Enk, *m/z* 554.2615, [M-H]-) for calibration. A mix containing 33 analytical standards of phenolic compounds (10 ppm) was prepared and injected in triplicate, prior to injection of the samples, to ensure the reproducibility of the instrument and confirm phenolic compound identification. In addition to the chemical standards, a set of quality control (QC) samples was also prepared by pooling equal volumes of each FVB extract. QC samples were injected after each batch of six runs of FVB samples to monitor the instrument’s stability.

MassLynx v 4.1 software (Waters, Milford, MA, USA) was used to acquire the MS data and Progenesis QI (Waters, Milford, MA, USA) software was applied to process the data. Metabolite identification was based on standard run parameters, such as: isotope distribution of neutral mass, exact mass, retention time, and MS/MS fragments spectra. Non-targeted identification was performed according to Sumner et al. [36] with a customized database built from PubChem and the online database Phenol-Explorer. The following parameters were applied in descending order of importance: exact mass error (<10 ppm); isotopic similarity (>80%); score (>30) and the highest score of fragmentation, generated by the software. Data from the literature and chemical characteristics of the molecules were also used to help the tentative identification of unknown compounds. In addition, only compounds present in the three technical replicates and showing CV < 30% were considered as tentatively identified. Finally, the relative amounts of phenolic compounds in each FVB sample were calculated by dividing the number of ions in a particular *m*/*z* ratio by the total number of ions detected.

### 2.5. Statistical Analysis

The data obtained were subjected to analysis of variance (one-way ANOVA) and means were compared using the Tukey and Bonferroni tests, when appropriate (95% confidence level, *p* < 0.05), using GraphPad Prism software version 5.0 (San Diego, CA, USA). Statistica software (Statsoft version 7.0, Hamburg, Germany) was used for elaboration and analysis of the experimental mixture planning. For multivariate data analysis, XLSTAT software (Addinsoft Inc., Ile-de-France, France) was applied for principal component analysis (PCA), while EZInfo 3.0 software (Waters) was applied for orthogonal partial least squares discriminant analysis (OPLS-DA).

## 3. Results and Discussion

### 3.1. Survey and Selection of Kitchen Scraps from Food Services

The FVB from the pre-preparation of two food services were quantified by a survey (weight in kg and volume in standardized 8.4 L bags) carried out for 20 days. Twenty-five different FVB were evaluated, seven of which were present in both locations. In Food Service A, the highest weight and volume were chard and carrots residues, which comprised the external apical part and peel, respectively (Appendix A). In Food Service B, the highest values were for potato (peels), papaya (apical part and seeds), and chayote residues (peels) (Appendix A). From the latter, only papaya and chayote were selected since they are less studied residues but may show bioactive potential due to their known high phenolic contents [37,38]. In total, five FVB were selected and processed as stated above (Section 2.1), including: carrot and chayote peels, papaya seeds, and the external apical part of papaya and chard, which have been named ‘papaya’ and ‘chard’ henceforth (Appendix A).

### 3.2. Extractor Selection by Mixing Planning

The statistical evaluation and critical values of extractors for each residue are shown in Table 1. The selection of the best extractor for each FVB was performed by analyzing the TRC and antioxidant capacity obtained by the DPPH method (Figure 1). Of the two results, the total reducing capacity (TRC) showed the greatest ability to distinguish the efficiency of the tested extractors. This may have been due to the lower solubility of the DPPH radical in very polar solvents such as water, which also provided lower reactivity of the radical with the antioxidant substance in the test [39,40]. In the case of two of the residues (chard and papaya seeds) neither test was able to differentiate between the mixtures of ethanol and water used, according to the statistical evaluation (Table 1). For each of the tests, this was credited to different reasons. For the TRC, all solvent combinations extracted similar concentrations of reducing compounds, with no significant difference between the tested mixtures. For the antioxidant capacity by DPPH, a limitation of the statistical analysis of this assay was demonstrated: the test is based on the premise that with an increase in the proportion of solvent in the mixture (ethanol or water, in this case), there will be an increased extraction of the desired compound until maximum extraction is achieved, from which point an increase in the proportion of solvent will lead to a reduction in the extracting power of the mixture. As linear or quadratic behaviors did not appear for these two residues, it was not possible to obtain a satisfactory answer from the applied statistical test.

The TRC results for carrot peel showed that there was no significant difference between the extractors with 50% and 75% ethanol, but the extractor with 50% ethanol stood out for its superiority in the DPPH antioxidant capacity analysis in relation to the other extractors (Figure 1). This could be explained by the rich carotenoid composition of carrot peel, since carotenoids are more soluble in nonpolar solvents [42].

For the chard residue (Figure 1), there was no significant difference between the extractors, except for 100% ethanol (six times less efficient than the other extractors). The DPPH analysis was then used to choose the extractor, where the extractors containing 25% and 100% water led to similar results. Thus, 100% water was selected as a solvent because it is environmentally friendly and low cost. In addition, the TRC values with the 100% water extractor were twice as high as those observed in the literature when methanol was applied to this same residue [43].

For the chayote peel, extraction with 100% water stood out as the most potent in obtaining phenolic compounds (746.46 ± 58.73 mg GAE/100 g) (Figure 1). Riviello-Flores et al. [44] also analyzed chayote extracts and identified four flavonols (rutin, myricetin, quercetin, and galangin), two dihydrocalones (phloretin and florizidine), and a flavanone (naringenin), which showed greater solubility in water and diluted alcohols, thus explaining the best solubility in this extractor.

The TRC values for papaya samples (Figure 1) were twice as high in the peels and apical part (1332.92 ± 14.28 mg GAE/100 g) compared to the seed (510.57 ± 43.71 mg GAE/100 g), which could be explained by the biosynthesis of polyphenols in the fruit peel for protection against solar radiation and as a defense mechanism against different types of stress during growth and ripening [41]. For the extraction of these compounds, the critical values indicated 60% water as the best extractor (Table 1). As these results differed from the experimental proportions tested, new assays were carried out with the following two proportions: 75:25 and 60:40 water:ethanol. These results showed no significant difference (Appendix A); therefore, the 75% water extractor was chosen for its greater potential as a biosolvent and lower cost. For papaya seed, the TRC values were not sufficient to differentiate the extractors since 25% to 100% water extractors presented similar TRC values (Figure 1). The 50% ethanol extractor was then chosen for this residue based on the DPPH analysis.

The extractors selected by mixture planning (SM) were: 100% water for chayote peel and chard; 75% water for papaya; and 50% ethanol for carrot peel and papaya seed. Finally, it is important to emphasize that the 80% methanol extractor, used for BPC extraction and applied as the control, did not present higher values for the five FVB, which supported the use of greener solvent mixtures and rendered the handling and disposal of extracts safer.

### 3.3. Selection of Ultrasonication Treatment Time

Among the treatment times evaluated, the results showed that the 30 min ultrasonic extraction method proved to be ideal for most residues in terms of TRC and DPPH values (Figure 2) and there was a decrease in extraction yield within 60 min. This behavior may be explained by the deeper diffusion of the solvent into the internal cell of the residues within a time of 30 min, thus allowing for extraction efficiency [45]; however, an additional increase in extraction time (to 1 h) may have cause degradation of some of the compounds, thus generating free radicals and depolymerization (Figure 2) [46]. The selected ultrasonication treatment time was, therefore, 30 min.

The effectiveness of the ultrasonication step in phenolic compound extraction may be observed in Table 2, where there was a significant increase in TRC values (US extract 2 times higher than SM extract), except for chayote peel where there was no significant difference between the two extracts. The significant increase in TRC after the application of ultrasonification may be credited to the collapse of cavitation bubbles near the cell walls of the plant matter, which helped break the cell walls and enhance contact between the solvent and solids. In addition, when the cavitation bubbles collapsed, an ultrasonic jet was produced that forced solvent into the cell to dissolve its components [47].

The antioxidant capacity by the DPPH method (Table 2) showed a significant increase in values, except for papaya residue with no significant difference and chard residue with a 24% reduction. Several reports confirmed that the DPPH method has been the most applied because it is considered more stable and sensitive in determining antioxidant capacity compared to other available methods, such as FRAP and ORAC [48]. DPPH has been used to determine the antioxidant capacity of leaf and fruit samples extracted in organic solvents such as n-hexane, acetone, ethyl acetate, and ethanol [49,50,51,52], thus demonstrating good performance in a variety of solvent polarities. However, it is also important to note that the phenolic compounds present in the evaluated plant matrices may exert other forms of antioxidant activities, in addition to free radical scavenging capacity, such as donating hydrogen atoms or electrons and chelating metal cations [53]. Therefore, for the present study, evaluation by other methods was also performed: FRAP and ORAC (Table 2). It was found that the DPPH analysis showed a strong correlation with FRAP (r = 0.89, *p* = 0.04) and moderate correlation with ORAC (r = 0.61, *p* = 0.27), the DPPH method being the most able to differentiate samples. The ORAC method showed similar results for all samples and, therefore, was not considered for the discussion of differences among extraction methods (Table 2).

It is very important to emphasize that regardless of the method, carrot peel and papaya showed the best results (172.51 ± 6.13 to 857.80 ± 51.66 µmol TE/g and 148.62 ± 9.54 to 389.88 ± 18.78 µmol TE/g, respectively). Furthermore, these residues had the highest flour yields (Table 2) and, therefore, would be the most suitable for future applications. Chayote peel and chard residue showed the lowest results among the samples (32.32 ± 2.24 to 192.22 ± 92.48 µmol TE/g and 46.28 ± 5.01 and 111.96 ± 47.81 µmol TE/g, respectively). Papaya seed showed intermediate results.

Even though it proved to be the least promising residue among all of the tested samples, the antioxidant capacity of chayote peel was 86% higher than the results reported in another study without ultrasonication application [54], highlighting again the effectiveness of this extraction method. This data, added to the relevance of the carrot peel phenolic compounds content, emphasized the promising bioactivity of this vegetable peel, which can boost the use and consumption of this integral vegetable and, consequently, reduce waste and increase bioactive phytochemical intake during meals.

The papaya apical part and peels showed higher antioxidant capacity than the seed of the same fruit (61% higher by the DPPH method and 2.6 times higher by FRAP analysis) (Table 2). This result supported the previously described hypothesis of the protective property of the fruit peel and, therefore, its higher phenolic content and bioactivity. Nevertheless, the papaya seed presented intermediate results among the samples, with an increase of 23% in the antioxidant capacity by the DPPH method after the application of ultrasonication. It is also important to highlight that the present study presented values ten times higher than those reported in the literature [55], thus demonstrating the importance of studying and optimizing the solvent mixture composition as well as auxiliary methods of extraction.

### 3.4. Phenolic Profile Evaluation in Each Extractor by ULPC-MS^E^

The SM, US, FPC, and BPC extracts of each of the 5 selected FVB were subjected to metabolomics analysis, totaling 20 extracts. Globally, 172 phenolic compounds were tentatively identified, 19 of which were confirmed with the injected commercial standards (please refer to Appendix A, compounds in bold). According to Figure 3A, the TPC extract (FPC + BPC) showed the highest number of identifications (105), followed by the SM (94) and US extracts (84). Moreover, 31 compounds were present in all samples. The superiority of the TPC extract may be explained by the sequential extraction applied, i.e., the alkaline and acid hydrolysis during the BPC extraction released 21 new compounds, which were crucial for the increased number of identifications in the TPC extract. Alkaline hydrolysis is effective in breaking ether and ester bonds, while acid hydrolysis breaks mainly glycosidic bonds; all of these bond types connect phenolic compounds to insoluble macromolecules in the cell matrix, e.g., cellulose, lignins, or hemicellulose. Therefore, the extraction of phenolic compounds was enhanced by sequential extraction compared to forms of extraction that released mostly free phenolic compounds [56,57].

The semi-quantification of the identified compounds by the total abundance of ions showed the US extract to have the highest abundance, i.e., 21% and 29% more abundant than in the TPC and SM extracts, respectively. Thus, US may be considered the best extractor to release higher amounts of phenolic compounds from these matrices (Figure 3B). Again, the TPC extract presented higher values than the SM extract, but, as in the number of identifications, the TPC extract was favored by the sequential extraction of alkaline/acid hydrolysis since the FPC fraction of the TPC extract (extraction like the others, in methodological terms) represented only 1/2 and 1/3 of the phenolic contents of the SM and US extracts, respectively, as may be observed in Figure 3B.

It was also possible to assign five different phenolic classes according to the quantification of the compounds identified in each extract: phenolic acids, flavonoids, other polyphenols, stilbenes, and lignans (Figure 3C). Both the SM and US extracts showed similar class distributions, indicating extracts with a similar phenolic profile; however, the application of the auxiliary method (US) increased phenolic acid extraction by 38%. This increase suppressed the flavonoid percentage relative to the phenolic acid percentage (Figure 3C). Nevertheless, flavonoids also showed a 19% increase in abundance after ultrasonication application. This increase indicated that this method may break the bonds linking them and increase the phenolic extraction rate, as these two classes are the main phenolic classes bound to the cell wall [56]. In contrast, the TPC extract presented the most dissimilar phenolic profile compared to the others, although the class of phenolic acids remained predominant, followed by other polyphenols and flavonoids. These findings were in agreement with the results of Do et al. [58] and Irakli et al. [59] that showed a relationship between the flavonoid content in plant samples and the extractor composition: different ethanol:water ratios were more efficient extractors than MeOH:water ratios, and the latter were not considered suitable for flavonoid extraction.

To further explore the variation in the dataset, multivariate analyses were applied. The PCA biplot (scores-samples; loadings-phenolic compounds) was used to investigate the degree of similarity or dissimilarity between extraction methods based on the profile of identified and semi-quantified compounds (Figure 4A). The PCA biplot indicated a clear distinction between extractors (PC1 and PC2 = 77%), showing that the phenolic content and profile were easily changed by the extraction used. Again, the TPC extract showed the greatest dissimilarity among samples, which directly impacted the high separation on the x-axis (PC1), while the SM and US extracts were only separated on the y-axis (PC2).

Finally, to characterize the phenolic compounds responsible for the extractor’s differentiation, OPLS-DA was also applied (Figure 5). The OPLS-DA model parameters were good in the comparison between the SM vs. US (Figure 5A; R2Y = 1, Q2 = 0.997), BPC vs. US (Figure 5B; R2Y = 1, Q2 = 0.999), and FPC vs. SM (Figure 5C; R2Y = 1, Q2 = 1) extracts, and the permutation test allowed to validate the prediction model (*p* = 0.05, 0.12 and 0.05, respectively). Thus, ten compounds were selected in each of the three analyses using the variable importance for the projection (VIP score, phenolic compounds in red, in Figure 5).

As shown in Figure 5A, it was possible to observe the impact of ultrasonication application on the samples, where seven phenolic compounds were more important (VIP score) in the US extract compared to the SM extract. Hence, two hypotheses may be formulated: (1) the application of ultrasonication favored the release of these compounds, which were previously bound to the cellular matrix, thereby improving their extractability; or (2) ultrasonication degraded the compounds present in the SM extract and formed new molecules. To test the second hypothesis, a new OPLS-DA was applied comparing BPC vs. US extracts (Figure 5B). Four US compounds selected by VIP in Figure 5A were repeated in Figure 5B (5-caffeoylquinic acid, dicaffeoylquinic acid isomer I, quercetin 3-O-glucuronide, and sinapic acid), indicating that they were phenolic degradation products. The increase in these phenolic acids after ultrasonication application (5-caffeoylquinic acid, dicaffeoylquinic acid isomer I, and sinapic acid) corroborated the results in Figure 3C (4% increase in phenolic acids between SM and US extracts) and supported the conclusion that these compounds were formed from the degradation of flavonoids (4% reduction in flavonoids between SM and US extracts). Furthermore, the flavonoid quercetin 3-O-glucuronide was formed by the degradation of the glycoside quercetin 3-O-rhamnosyl-rhamnosyl-glucoside, which was only present in the SM extract.

The other compounds present only in the US extract (Figure 5B, hispidulin and apigenin 7-O-apiosyl-glucoside) were not found in the comparison between the SM and US extracts (Figure 5A), but only in the comparison between the BPC and US extracts. This supported the hypothesis that these two compounds were unique to the US extract due to the applied extractor (ethanol) and not the processing (ultrasonication). To confirm this hypothesis, a new OPLS-DA was applied comparing FPC vs. SM extracts (Figure 5C), i.e., a comparison only between methanol and ethanol. The two compounds (hispidulin and apigenin 7-O-apiosyl-glucoside) were found exclusively in the SM extract, confirming that they were best extracted in ethanol. Thus, Figure 5C, together with the data previously presented in Figure 3A, indicated that the use of ethanol favored the extraction of flavonoids (28% less flavonoid abundance in the TPC extract compared to the SM extract).

### 3.5. Metabolomics Characterization of Each FVB by UHPLC-MS^E^

In addition to the differences between extractors, the characterization of each FVB was also carried out. Chard stood out for the highest number of identifications (90 phenolic compounds), followed by papaya (83), chayote peel (73), papaya seed (71), and carrot peel (51). It is important to highlight that one compound may be present in two or more samples and, furthermore, 16 phenolic compounds were common among the five FVB analyzed in this study (Figure 3A).

The abundance data verified that different FVB presented better results for different types of extraction (Figure 3B): US extraction was more suitable for carrot peel and papaya samples, sequential extraction (TPC) was more suitable for chard and papaya seed, and chayote peel showed no significant difference between SM and sequential extraction (TPC). Only chayote showed a reduction in abundance values after ultrasonication application (36% reduction compared to SM). According to ref. [60], the application of ultrasonication was effective in PC extraction, but prolonged treatments with high frequency (358 and 850 kHz) and/or high force (750 W) could degrade/oxidize PC. The hypothesis for the present study was that, due to the phenolic composition of chayote, the ultrasonication time applied (30 min) may have been too long, thus degrading the compounds. In the future, new intermediate times should be tested for this sample.

Among the different classes of phenolic compounds, phenolic acids were predominant in carrot peel, papaya seed, and chard (98%, 58%, and 44%, respectively), followed by flavonoids (1%, 22%, and 38%, respectively). The highest ionic intensity for phenolic acids agreed with the results reported by Peng, Li, Li, Deng and Zhang [56] in which this class stood out regardless of the extractor, as shown in Figure 3C. However, when the number of identifications was considered for these same samples, flavonoids stood out (3/7 of total identifications), followed by phenolic acids (1/3 of total identifications) (Appendix A).

In contrast, flavonoids were more abundant in chayote peel and papaya (67% and 52%, respectively), followed by phenolic acids (29% and 43%). Other studies with papaya residues also identified higher values for flavonoids compared to phenolic acids [55,61]. Vieira, Pinho, Ferreira and Delerue-Matos [37] analyzed different parts of chayote and observed a greater presence of flavonoids in the peel (approximately 2 times higher) compared to the pulp; a fact later confirmed by Kuppusamy et al. [62], where FVB peels had higher amounts of flavonoids among twenty-one plant residues. It is also important to highlight that the class of other polyphenols (e.g., pyrogallol, 4-hydroxybenzaldehyde, vanillin) was the third most abundant class in all samples, and that lignans and stilbenes presented values close to 0%.

Among the 172 phenolic compounds tentatively identified, the ten most abundant of each FVB were selected, as shown in Table 3. The ten most abundant compounds in carrots included five derivatives of chlorogenic acid (isomers of caffeoylquinic and dicaffeoylquinic acid) and one derivative of ferulic acid (3-feruloylquinic acid), corroborating the results reported by Shahidi et al. [63]. It is important to highlight that the compounds mentioned above were associated with antioxidant activity [64] and that their contents were higher in carrot peels than in the inner parts (phloem and xylem) and can, therefore, be considered a promising source for the production of value-added products.

Like carrot peel, chard also had 3-feruloylquinic acid as one of its ten most abundant compounds (8th) and mostly presented flavonoids (Table 3: 1st, 7th, and 9th most abundant compounds) as well as hydroxycinnamic acids and their derivatives (Table 3: 2nd, 4th, and 8th most abundant compounds). Although some compounds previously reported in other studies were not identified, such as myricetin 3-O-rhamnoside (myricitrin) [65], its glycosylated isobar was the ninth most abundant in the present sample. The most reported flavonoid in chard [66], 2″-xylosylvitexin, was not found in its residue.

Globally, papaya residue, and chayote peel each had 5 flavonoids, 4 phenolic acids, and 1 other polyphenol among the ten most abundant compounds (Table 3). Flavonoids are widely reported for their antioxidant activities and pharmacological properties [37], highlighting again the importance of using residues rich in this class of phenolic compounds. Like chard, papaya presented quercetin 3-O-glucuronide and sinapic acid as the first and second most abundant compounds, respectively. In addition, these two samples shared other four compounds, but in different rankings (4-hydroxybenzaldehyde, ferulic acid, epigallocatechin, and 3-feruloylquinic acid), showing a similarity between them that may be explained by the fact that both FVB (chard and papaya) were generated by the discarded part closest to the peduncle of the plant. Conversely, chayote did not show similarity with the other samples, as 70% of its most abundant compounds were unique; however, the present identification was similar to that reported by Diaz de Cerio et al. [67], i.e., the presence of derivatives of apigenin and chrysoeriol.

In papaya seed, kaempferol was the most abundant compound (Table 3). This compound also appeared as the tenth most abundant compound in the papaya apical part and peel, supporting the conclusion that this is a characteristic compound of this fruit. The study carried out by Rodrigues, Mazzutti, Vitali, Micke and Ferreira [38] also identified kaempferol as part of the phenolic composition of papaya seed, but with undetectable amounts. Also, according to these authors, gallic acid and scopoletin were identified for the first time in papaya seed, but with low amounts. These compounds were the fourth and eighth most abundant compounds in the present study, respectively. These results showed the efficiency of the extraction methods developed and the analytical tools applied that provided evidence of previously unidentified phenolic compounds or those found only in negligible amounts.

In addition, two different PCAs were performed: (a) with the distribution of FVB regardless of the extractor used (Figure 4B); and (b) with the distribution of extractors in each FVB (Appendix A). Although the general PCA shown in Figure 4B was not as effective in observing the variability of the samples (PC = 60%), the proximity of some FVB was evident. The papaya seed showed great dissimilarity among the other samples, standing out as the main component responsible for separation of the y axis (Figure 4B, 27%). The residues from the peduncle region (chard and papaya) were more closely distributed in Figure 4B (in the lower-left quadrant). This similarity was also reported above, with the presence of some shared compounds among the most abundant (Table 3). This behavior could lead to the hypothesis that the phenolic profile of the residues was influenced by the layer of vegetable/fruit that was removed, regardless of the type of FVB. However, this behavior was not observed in the peels (chayote and carrot), where carrot was centered in the graph and chayote appeared in the upper-right quadrant (Figure 4B).

Finally, Appendix A showed the great variability between extractors (PC value between 89 and 99%). In all FVB, the SM and US extracts were aligned on the x-axis, separated only by the y axis, while the TPC extract was more centralized to the y-axis and exclusively separated by the x-axis. These data corroborated the alignment between these two extractors already pointed out in Figure 4A. These results indicated that the extractor also plays a major role, since its behavior is similar regardless of the analyzed FVB.

## 4. Conclusions

The selection of the main food residues in two food services, as well as their quantification, was important to show the dimension of food waste and evaluate the matrices that can be reused. The parts of vegetables often considered inedible (peel, stalk, and seeds) presented, in general, a rich composition of bioactive compounds responsible for several biological activities, which were highlighted by the antioxidant activities explored in this study. The planning of mixtures was essential in the extraction of these compounds, using solvents considered safe (ethanol and water, in the ideal proportions for each FVB) and favored by the physical ultrasonication process (30 min).

The metabolomics approach revealed the phenolic profile of each residue and, together with the multivariate analyses, indicated similarities between FVB extracted from the same parts of the vegetable/fruit, i.e., peels and stalks showed similarities to each other. Furthermore, metabolomics revealed that the identification/quantification of these compounds was influenced by the applied extractor and demonstrated the advantage of SM and US compared to the control (TPC), mainly in the extraction of flavonoids.

Knowledge about the rich phenolic composition of these fruit and vegetable by-products and their extraction, provided by the present study, makes it possible to explore and valorize these extracts as alternative raw materials in order to obtain high added-value products such as functional or nutraceutical ingredients. In addition, the study promotes the whole use of these foods, thus helping to reduce the environmental impact of food waste. 

## Figures and Tables

**Figure 1 metabolites-13-00386-f001:**
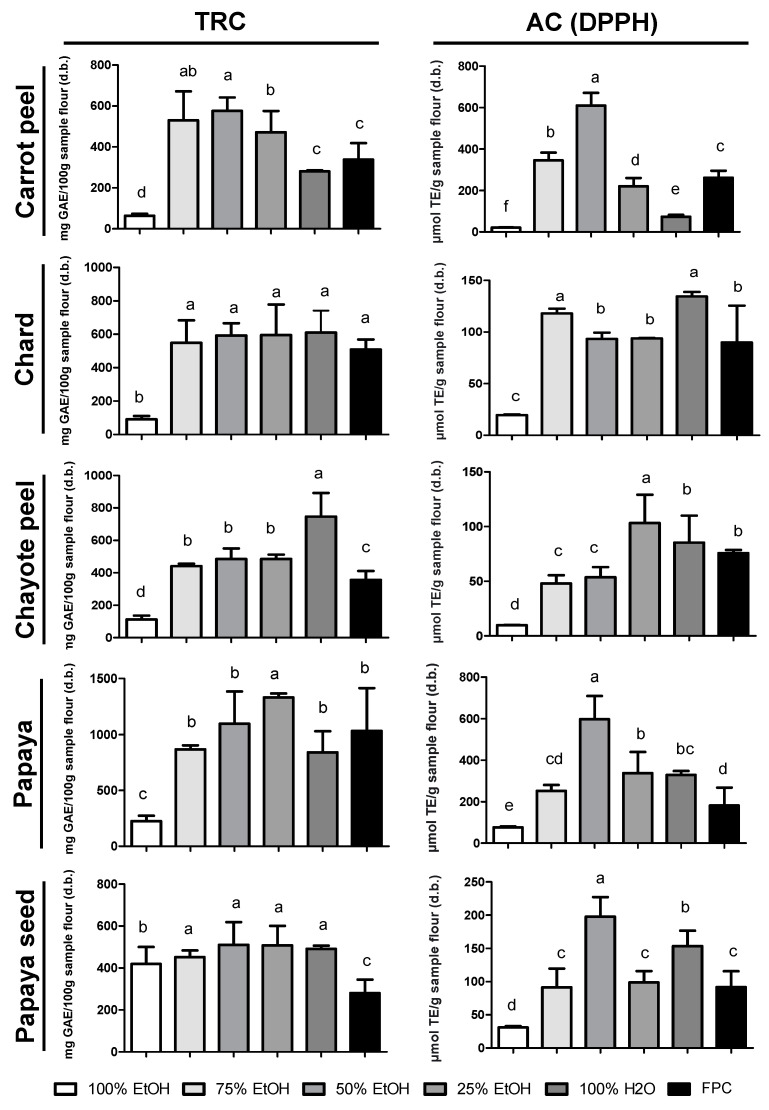
Evaluation of total reducing capacity (TRC) and antioxidant capacity (AC) by the DPPH method using six different extractors on five fruit and vegetable by-products (FVB). Results are expressed as mean ± standard deviation (*n* = 3). Different letters indicate a significant difference between extractors (Tukey, *p* < 0.05).

**Figure 2 metabolites-13-00386-f002:**
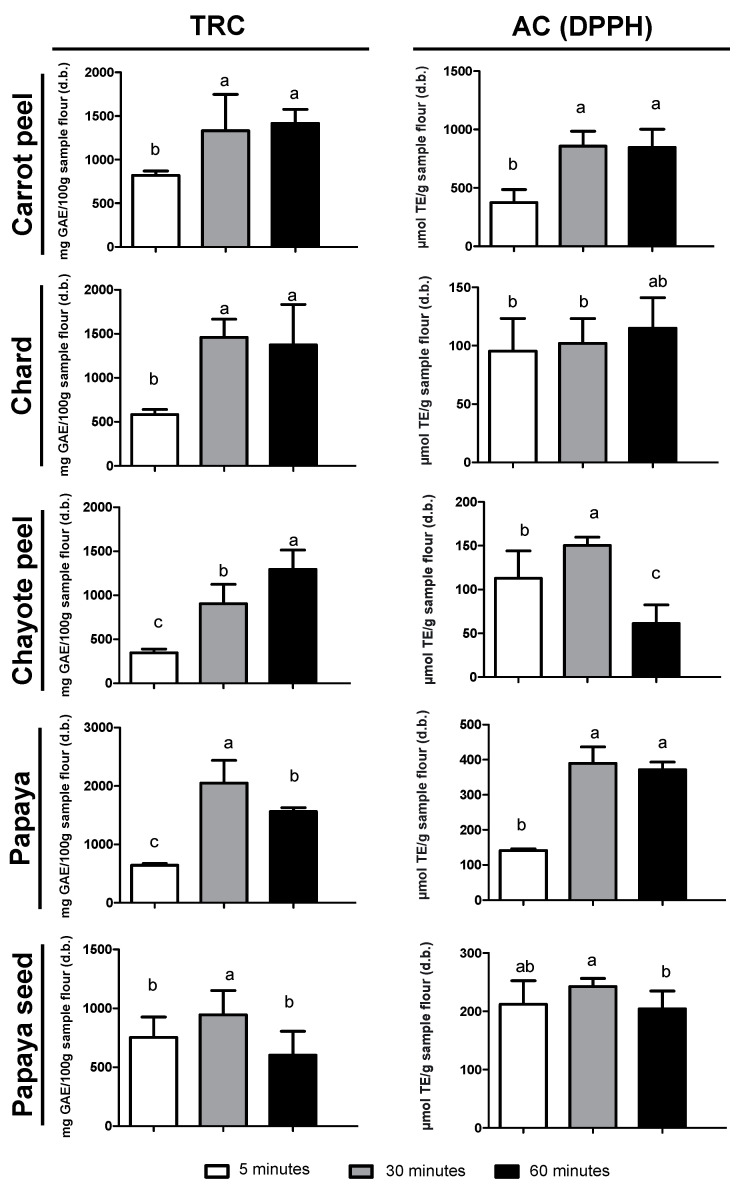
Evaluation of ultrasonication (US) extraction for 3 different times in terms of total reducing capacity (TRC) and antioxidant capacity (AC) by the DPPH method on five fruit and vegetable by-products (FVB). Results are expressed as mean ± standard deviation (*n* = 3). Different letters indicate a significant difference between US time (Tukey, *p* < 0.05).

**Figure 3 metabolites-13-00386-f003:**
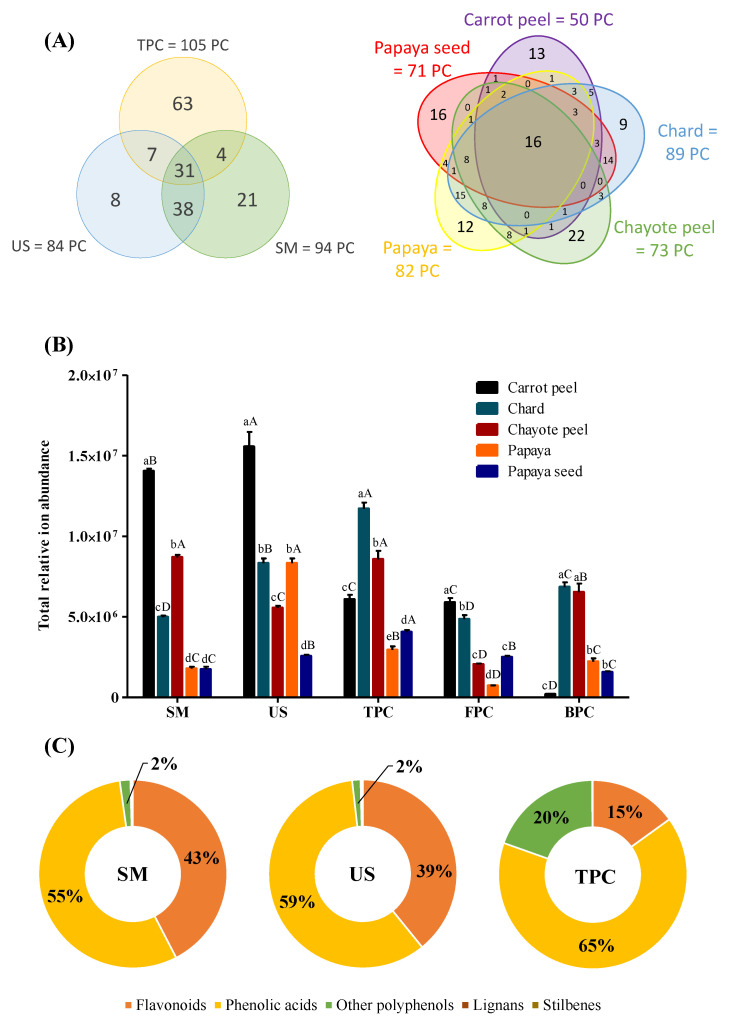
Metabolomics analysis: (**A**) Venn diagram with the number of identifications distribution in each extract and in each of the fruit and vegetable by-products (FVB); (**B**) total relative ion abundance of phenolic compounds with the sum (Σ) of each extract; (**C**) distribution of phenolic classes in selected mixture (SM), ultrasonication (US) and total phenolic content (TPC) extracts. Different lowercase letters mean a significant difference (*p* < 0.05) between FVB with the same extractor, while uppercase letters mean a significant difference (*p* < 0.05) between extracts from the same FVB. Bars represent standard deviation (*n* = 3).

**Figure 4 metabolites-13-00386-f004:**
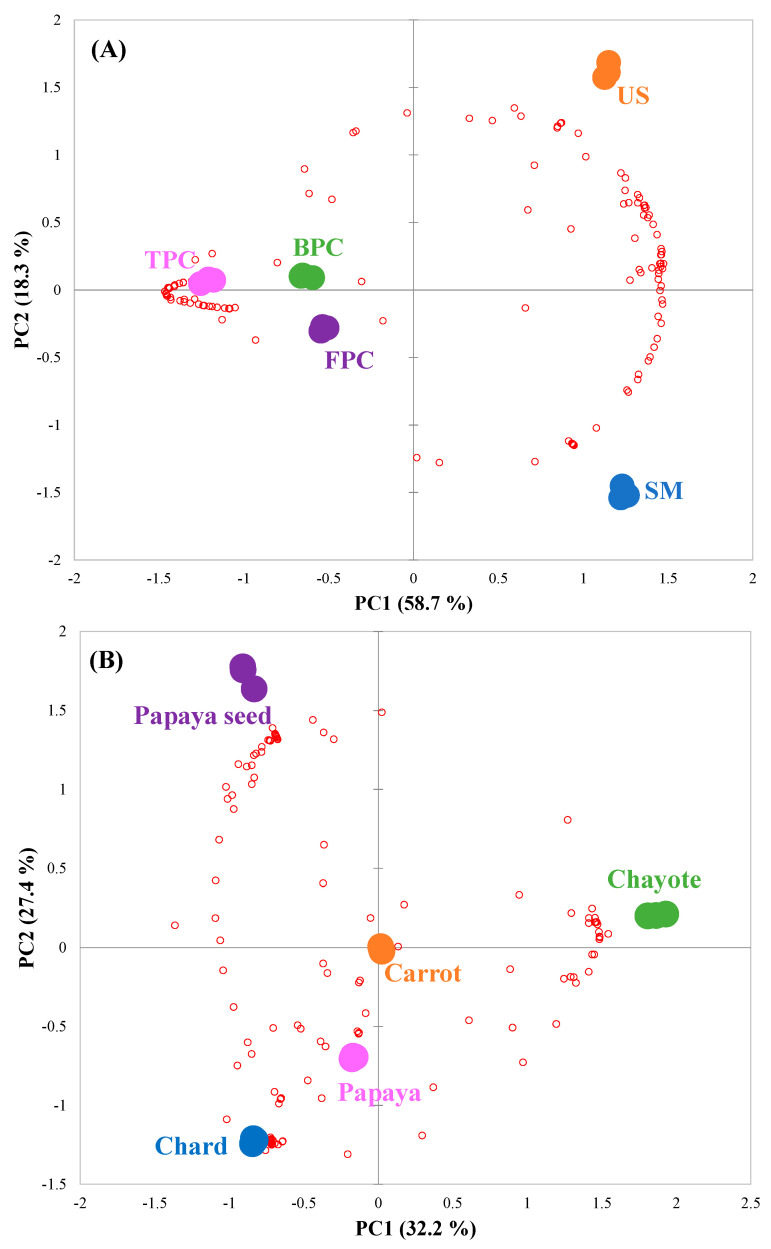
Principal component analysis (PCA) biplot to observe the similarity/dissimilarity between (**A**) the extract used, independent of the fruit and vegetable by-products (FVB), and (**B**) the fruit and vegetable by-products (FVB), regardless of the extract. The samples (symbols) are distributed according to relative intensity of identified phenolic compounds (red circles).

**Figure 5 metabolites-13-00386-f005:**
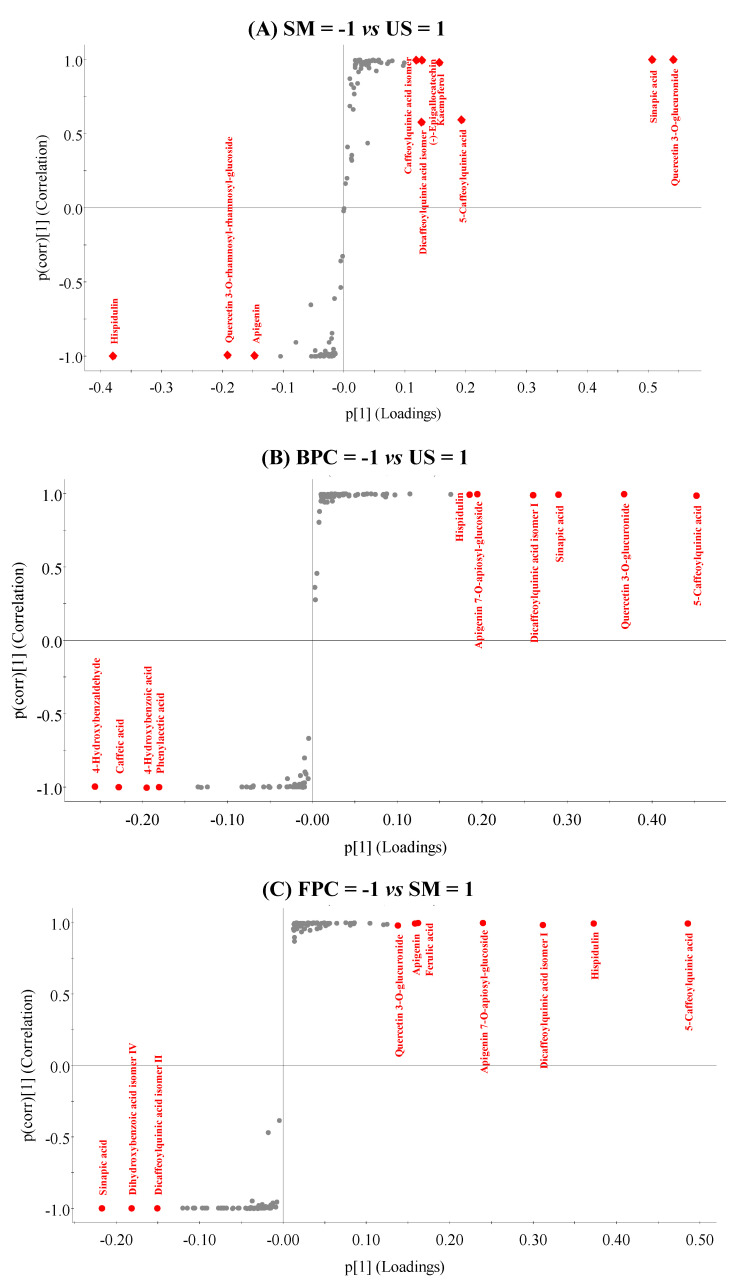
Comparative analysis between extracts: s-plot of orthogonal partial least squares discriminant analysis (OPLS-DA) of (**A**) ultrasonication (US) vs. total phenolic content (TPC) extracts, (**B**) selected mixture (SM) vs. US extracts, and (**C**) TPC vs. SM extracts.

**Table 1 metabolites-13-00386-t001:** Analysis of variance [41] for the extracts of the five fruit and vegetable by-products (FVB), considering the total reducing capacity (TRC) and antioxidant capacity (AC) by the DPPH method.

Sample	Statistical Model	Sum of Squares	df	F-Value	*p*-Value	R^2^	Critical Value	
							EtOH (%)	H_2_O (%)
**TRC**								
**Carrot peel**	Linear	15,639.20	1	0.26	0.65	0.08	46	54
	Quadratic	170,265.40 *	1	26.04 *	0.04 *	0.93		
**Chard**	Linear	118,045.90	1	4.37	0.13	0.59	30	70
	Quadratic	61,589.00	1	6.30	0.13	0.90		
**Chayote peel**	Linear	172,731.10 *	1	16.12 *	0.03 *	0.84 *	0	100
	Quadratic	2349.40	1	0.16	0.73	0.86		
**Papaya**	Linear	308,433.00	1	2.19	0.24	0.42	37	63
	Quadratic	391,315.20 *	1	25.48 *	0.04 *	0.96 *		
**Papaya seed**	Linear	4480.59	1	5.44	0.10	0.64	28	72
	Quadratic	2034.99	1	9.30	0.09	0.94		
**AC (DPPH)**								
**Carrot peel**	Linear	47.80	1	0.00	0.98	0.00	50	50
	Quadratic	202,871.40	1	8.93	0.10	0.82		
**Chard**	Linear	4228.72	1	3.62	0.15	0.55	10	90
	Quadratic	586.27	1	0.40	0.59	0.62		
**Chayote peel**	Linear	4254.19 *	1	13.27 *	0.04 *	0.82 *	0	100
	Quadratic	333.33	1	1.06	0.41	0.88		
**Papaya**	Linear	37,539.00	1	0.98	0.39	0.25	39	61
	Quadratic	72,929.00	1	3.52	0.20	0.73		
**Papaya seed**	Linear	6998.84	1	1.95	0.26	0.40	30	70
	Quadratic	3672.29	1	1.03	0.42	0.60		

Data with “*” are significant values (*p* < 0.05). df = degrees of freedom. TRC—total reducing capacity; AC—antioxidant capacity; DPPH (2-diphenyl-1-picrylhydrazyl).

**Table 2 metabolites-13-00386-t002:** Comparison between the selected mixture extract (SM) and ultrasonication extract (US) of five fruit and vegetable by-products (FVB) in terms of total reducing capacity (TRC) and antioxidant capacity (AC) by DPPH method as well as the antioxidant capacity of ultrasonication extracts by FRAP and ORAC methods.

Sample	Extract	TRC (mg GAE 100 g^−1^)	AC (DPPH) (µmol TE g^−1^)	AC (FRAP) (µmol FeSO_4_ g^−1^)	AC (ORAC) (µmol TE g^−1^)
**Carrot peel**	SM	576.83 ± 26.20 ^b^	610.62 ± 24.44 ^b^	172.52 ± 6.13 ^A^	214.01 ± 92.47 ^A^
	US	1332.95 ± 166.82 ^a^	857.80 ± 51.66 ^aA^		
**Chard**	SM	610.41 ± 52.98 ^b^	134.43 ± 1.78 ^a^	46.28 ± 5.01 ^BC^	111.96 ± 47.81 ^A^
	US	1461.56 ± 82.43 ^a^	102.10 ± 8.50 ^bD^		
**Chayote peel**	SM	746.46 ± 58.73 ^a^	85.26 ± 9.97 ^b^	32.32 ± 2.24 ^C^	192.22 ± 92.48 ^A^
	US	903.55 ± 89.00 ^a^	150.21 ± 3.84 ^aD^		
**Papaya**	SM	1332.92 ± 14.28 ^b^	338.30 ± 41.15 ^a^	148.62 ± 9.54 ^A^	225.18 ± 97.29 ^A^
	US	2048.63 ± 157.34 ^a^	389.88 ± 18.78 ^aB^		
**Papaya seed**	SM	510.57 ± 43.71 ^b^	197.61 ± 11.94 ^b^	57.46 ± 5.62 ^B^	183.59 ± 122.57 ^A^
	US	945.90 ± 82.81 ^a^	242.66 ± 5.49 ^aC^		

Results are expressed per g or 100 g in dry basis as mean ± standard deviation (*n* = 3). Different letters mean a significant difference (*p* < 0.05) between the SM and US of each sample (lower case); and between the ultrasonication residues in antioxidant capacity analyses (upper case). SM (selected mixture), US (ultrasonication extract) GAE (gallic acid equivalent), TE (Trolox equivalent), FeSO_4_ (ferrous sulphate), TRC (total reducing capacity), AC (antioxidant capacity), DPPH (2-diphenyl-1-picrylhydrazyl), FRAP (ferric reducing antioxidant power), and ORAC (oxygen radical absorbance capacity).

**Table 3 metabolites-13-00386-t003:** List of the ten most abundant compounds (in descending order) present in the fruit and vegetable by-products (FVB).

Possible Identifications	*m/z* (exp)	RT (min)	Molecular Formula	Score (%)	FS (%)	Mass Error (ppm)	IS (%)	Class
**Carrot peel**								
**5-caffeoylquinic acid**	**353.0864**	**3.41**	**C_16_H_18_O_9_**	**45.1**	**33.3**	**−3.91**	**96.87**	**PA**
Dicaffeoylquinic acid isomer IV	515.1186	6.29	C_25_H_24_O_12_	44.1	25.0	−1.77	97.62	PA
Dicaffeoylquinic acid isomer II	515.1200	5.66	C_25_H_24_O_12_	41.1	8.2	0.90	98.21	PA
3-feruloylquinic acid	367.1025	3.59	C_17_H_20_O_9_	45.4	31.2	−2.69	98.92	PA
Dicaffeoylquinic acid isomer III	515.1199	6.03	C_25_H_24_O_12_	50.7	56.8	0.74	97.37	PA
Caffeoylquinic acid isomer II	353.0870	2.68	C_16_H_18_O_9_	42.8	21.6	−2.41	95.10	PA
**Benzoic acid**	**121.0285**	**3.35**	**C_7_H_6_O_2_**	**38.0**	**0**	**−8.12**	**99.23**	**PA**
5-tricosylresorcinol	431.3867	3.26	C_29_H_52_O_2_	36.9	8.5	−6.33	83.42	OP
**Vanillic acid**	**167.0342**	**2.03**	**C_8_H_8_O_4_**	**53.0**	**72.7**	**−4.57**	**97.69**	**PA**
3,4-dihydroxyphenylacetic acid	167.0352	2.93	C_8_H_8_O_4_	58.6	96.1	1.13	98.08	PA
**Chard**								
Quercetin 3-O-glucuronide	477.0632	4.66	C_21_H_18_O_13_	41.2	25.7	−8.84	90.24	F
**Sinapic acid**	**223.0598**	**5.56**	**C_11_H_12_O_5_**	**46.1**	**40.2**	**−6.35**	**97.63**	**PA**
**4-hydroxybenzaldehyde**	**121.0290**	**7.55**	**C_7_H_6_O_2_**	**57.8**	**94.9**	**−3.91**	**98.89**	**OP**
Ferulic acid	193.0490	5.53	C_10_H_10_O_4_	42.0	21.7	−8.59	97.67	PA
**4-hydroxybenzoic acid**	**137.0238**	**5.92**	**C_7_H_6_O_3_**	**50.5**	**58.1**	**−4.30**	**99.49**	**PA**
**2,5-dihydroxybenzoic acid**	**153.0185**	**3.10**	**C_7_H_6_O_4_**	**38.6**	**0**	**−5.27**	**99.01**	**PA**
**(-)-epigallocatechin**	**305.0690**	**3.80**	**C_15_H_14_O_7_**	**38.9**	**7.3**	**7.47**	**95.41**	**F**
3-feruloylquinic acid	367.1025	3.59	C_17_H_20_O_9_	45.4	31.2	−2.69	98.92	PA
Myricetin 3-O-glucoside	479.0828	5.93	C_21_H_20_O_13_	42.6	16.7	−0.58	97.15	F
Protocatechuic aldehyde	137.0237	2.93	C_7_H_6_O_3_	55.2	83.6	−5.47	98.84	OP
**Chayote peel**								
Hispidulin	299.0545	9.21	C_16_H_12_O_6_	50.1	59.2	−5.43	97.33	F
Apigenin 7-O-apiosyl-glucoside	563.1400	4.60	C_26_H_28_O_14_	51.0	62.0	−1.07	94.09	F
**Caffeic acid**	**179.0346**	**7.67**	**C_9_H_8_O_4_**	**56.8**	**87.7**	**−1.94**	**98.56**	**PA**
Phenylacetic acid	135.0448	3.60	C_8_H_8_O_2_	39.3	0	−2.55	99.76	PA
**4-hydroxybenzoic acid**	**137.0238**	**5.92**	**C_7_H_6_O_3_**	**50.5**	**58.1**	**−4.30**	**99.49**	**PA**
Apigenin	269.0440	8.89	C_15_H_10_O_5_	39.5	8.2	−5.82	96.13	F
Chrysoeriol 7-O-apiosyl-glucoside	593.1509	4.10	C_27_H_30_O_15_	47.8	43.4	−0.52	96.46	F
Protocatechuic aldehyde	137.0237	2.93	C_7_H_6_O_3_	55.2	83.6	−5.47	98.84	OP
Neohesperidin	609.1879	0.57	C_28_H_34_O_15_	43.3	37.1	8.88	89.46	F
Dihydroxybenzoic acid isomer IV	153.0189	3.64	C_7_H_6_O_4_	39.1	0	−3.04	98.92	PA
**Papaya**								
Quercetin 3-O-glucuronide	477.0632	4.66	C_21_H_18_O_13_	41.2	25.7	−8.84	90.24	F
**Sinapic acid**	**223.0598**	**5.56**	**C_11_H_12_O_5_**	**46.1**	**40.2**	**−6.35**	**97.63**	**PA**
Ferulic acid	193.0490	5.53	C_10_H_10_O_4_	42.0	21.7	−8.59	97.67	PA
Quercetin 3-O-rhamnosyl-rhamnosyl-glucoside	755.2042	4.55	C_33_H_40_O_20_	48.5	54.0	0.25	88.75	F
**(-)-epigallocatechin**	**305.0690**	**3.80**	**C_15_H_14_O_7_**	**38.9**	**7.3**	**7.47**	**95.41**	**F**
**4-hydroxybenzaldehyde**	**121.0290**	**7.55**	**C_7_H_6_O_2_**	**57.8**	**94.9**	**−3.91**	**98.89**	**OP**
Diosmetin 7-O-rutinoside	607.1665	6.14	C_28_H_32_O_15_	38.3	3.6	−0.50	88.71	F
**Benzoic acid**	**121.0285**	3.35	**C_7_H_6_O_2_**	**38.0**	**0**	**−8.12**	**99.23**	**PA**
3-feruloylquinic acid	367.1025	3.59	C_17_H_20_O_9_	45.4	31.2	−2.69	98.92	PA
**Kaempferol**	**285.0391**	6.40	**C_15_H_10_O_6_**	**38.4**	**0**	**−4.87**	**97.40**	**F**
**Papaya seed**								
**Kaempferol**	**285.0391**	**6.40**	**C_15_H_10_O_6_**	**38.4**	**0**	**−4.87**	**97.40**	**F**
Phenacetylglycine	192.0648	3.72	C_10_H_11_NO_3_	37.6	0	−9.48	98.31	PA
Dihydroxybenzoic acid isomer IV	153.0189	3.64	C_7_H_6_O_4_	39.1	0	−3.04	98.92	PA
**Gallic acid**	**169.0133**	**2.50**	**C_7_H_6_O_5_**	**48.4**	**49.2**	**−5.32**	**99.03**	**PA**
O-Methylgallic acid isomer II	183.0281	4.20	C_8_H_8_O_5_	37.5	0	−10.00	98.50	PA
**Benzoic acid**	**121.0285**	**3.35**	**C_7_H_6_O_2_**	**38.0**	**0**	**−8.12**	**99.23**	**PA**
**4-hydroxybenzoic acid**	**137.0238**	**5.92**	**C_7_H_6_O_3_**	**50.5**	**58.1**	**−4.30**	**99.49**	**PA**
Scopoletin	191.0332	3.60	C_10_H_8_O_4_	40.7	14.3	−9.36	99.45	OP
**4-hydroxybenzaldehyde**	**121.0290**	**7.55**	**C_7_H_6_O_2_**	**57.8**	**94.9**	**−3.91**	**98.89**	**OP**
Protocatechuic aldehyde	137.0237	2.93	C_7_H_6_O_3_	55.2	83.6	−5.47	98.84	OP

*m*/*z* = mass/load; RT = retention time; FS = fragmentation score; IS = isotopic similarity; PA = phenolic acids; F = flavonoids; OP = other polyphenols. Compounds in bold represent reference standards.

## Data Availability

Data is not publicly available due to privacy or ethical restrictions.

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
