# Peer review of "Food Service Kitchen Scraps as a Source of Bioactive Phytochemicals: Disposal Survey, Optimized Extraction, Metabolomic Screening and Chemometric Evaluation"

_metabolites, 2023, doi:10.3390/metabo13030386_

Round 1

Author Response

Dear Reviewer 1,

Thank you for the opportunity to revise our paper “Food Service’s Kitchen scraps as a source of bioactive phytochemicals: disposal survey; optimized extraction, metabolomic screening and chemometric evaluation”. We appreciate the time and effort dedicated by the reviewers to providing your valuable feedback on our manuscript, and all the suggestions offered have been immensely helpful.

We have included the reviewer comments immediately after this letter and they were answered point-by-point (blue color). All the remarks have been considered and the new version of the manuscript has been included. The corresponding changes can be visualized on “Tracked Changes” and I further declare that the revisions have been approved by all authors.

We hope the revised manuscript can be published in Metabolites, and we thank you for your continued interest in our research.

After I read the article, I concluded that the article is well-organized, contain the components: Front matter (Title, Author list, Affiliations, Abstract, Keywords), Research manuscript sections (Introduction, Materials and Methods, Results and Discussion, Conclusions), Back matter (Acknowledgements and funding, References). All the sections are well-developed, the article well-written and easy to understand, the authors do a good job of synthesizing the literature. I consider that the article is very important to the development of knowledge in this particular area of interest. Below are the main points that I think should be reviewed and my comments.

Comment 1.    Lines 107-108: I suggest to the authors to provide the type and provenience of the instruments (stirrer and centrifuge).

Response: This is a relevant observation, and we appreciate it. This information was included in the revised manuscript.

Comment 2.  Lines 109-111:  I suggest to the authors to provide information about how the total reducing capacity (TRC) is obtained by the Folin-Ciocalteu method.

Response: We appreciate the suggestion. In this case, the Folin-Ciocalteu and DPPH methods are described in topic 2.3 and we believe that describing them in topic 2.2 would make the manuscript repetitive. As we agree that this information may be confusing for readers, we have included the sentence “both described in topic 2.3” in parentheses to direct the reader to the description of how the analysis was performed.

Comment 3.  Lines 133-147:    For the determination of the antioxidant capacity, I suggest the authors to provide a brief presentation of the DPPH, FRAP and ORAC methods (sample preparation method, reagents used - amounts and concentrations, working method and calculation equations).

Response: We appreciate the comment and agree that this information was missing from the initial manuscript. In order to make the execution of the analysis clearer, we rewrote the entire topic 2.3 of the manuscript, making it more descriptive.

Comment 4. Lines 153-154: The composition of the mobile phase must be provided. What represent component “A” and what is the component “B”?

Response: Thank you for this remark. Mobile phase A consists of ultra-pure water containing 0.3% formic acid and 5 mM ammonium formate, while mobile phase B is acetonitrile containing 0.3% formic acid. We agree that this important information is missing from the manuscript, and we have already included it in the revised manuscript.

Reviewer 2 Report

Manuscript title: Food Service’s kitchen scraps as a source of bioactive phytochemicals: disposal survey; optimized extraction, metabolomic screening and chemometric evaluation  

This study has certain significance in bioactive phytochemicals  in food services... However, revisions are necessary for the current version of the manuscript. The following questions to be addressed/considered may be helpful to improve the manuscript.

Major comments

·       Full detail of the author’s affiliation is needed, the institute, city, postal code and the country

·       Insufficient Abstract: In the abstract, the main aim and background of the manuscript are missing, the current version it only highlights the result. In addition, it would be even better to have a sentence as a future perspective.

·       The unit/abbreviation is not mentioned before, consider defining the abbreviation when mentioned for the first time…. Please check throughout the manuscript to define the abbreviations.

·       Lake of scientific literature to support the statements and findings throughout the manuscript…... I have made some suggestions for that and more need it….

·       Grammar and punctuation issuers need to be addressed. I have selected/mentioned some as examples.

·       I have a major concern about the results and discussion section. The authors describe the results and compare the results with previous studies, however, insight mechanisms are still insufficient.

Specific comments:

Abstract

If the unit/abbreviation is not mentioned before, consider defining the abbreviation when mentioned for the first time.

Line 20: replace ‘’ultrasound’’ with ultrasonication’’

And with ‘’class’’ do you mean ‘’compounds’’, as I see in the text you’ve discussed just flavonoids, not other classes of flavonoids??

Line 21-23: A complicated sentence, please revise

Introduction:

Line 40: A reference is needed here, for example, you can use:

https://doi.org/10.3945/an.112.002154

Gupta, V.K., Tuohy, M.G., O'Donovan, A. and Lohani, M. eds., 2015. Biotechnology of bioactive compounds: sources and applications. John Wiley & Sons.

Line 58-63: A complicated sentence, please revise and check the grammar

Line 59: A reference is needed here, for example:

https://doi.org/10.1016/j.foodres.2020.109854

Line 71: A reference is needed here, for example, you can use:

https://doi.org/10.1080/22297928.2016.1152912

Line 75-78: These are rather long sentences, better to break them down into more sentences.

Line 62-66: A complicated sentence, please revise and check the grammar

In MM section

Literature references are missing for all sub-section. It would be better to cite the references that the procedure adopted.

Additional info is needed for the table caption, most importantly significant figures.

In MM section, what is the quality control (QC) data? There is no mention of the QC.

What is the accuracy of the instruments, recovery, LOD, and LOQ ……. These parameters are needed to report the efficiency of any analytical system.

In general, how many times you’ve recorded the data,? duplicate? Triplicate?..... what you mentioned in the text is not clear, please elaborate more on this

R&D section

Table 1: Consider to have harmonize the significant figures for all variables.

What is the difference between Table 3 and table S1? If the data ais the same consider deleting one, you don’t have to preset the data twice, despite the table 3 is summarized.

These sections are repeating information already presented and explain things in an unnecessarily complicated way. The quality of the manuscript would benefit from the whole section being condensed, Line 236-260, Line 293-333, Line 409-433, Line 492-526…..

Line 197-198: even though you have defined the abbreviation, nut you still repeated the full name, consider using the abbreviation you define. Please check throughout the manuscript.

Figures 1, and 2. how the comparing made between the treatments and assigning the letter for the statistical difference is confusing. For example, Figure 1 a: How you can have d, ab, a, b, c, c, …..? Please elaborate more!

Conclusion

Important conclusions! However, the future perspectives for the following research are highly crucial here …..

Author Response

Dear Reviewer 2,

Thank you for the opportunity to revise our paper “Food Service’s Kitchen scraps as a source of bioactive phytochemicals: disposal survey; optimized extraction, metabolomic screening and chemometric evaluation”. We appreciate the time and effort dedicated by the reviewers to providing your valuable feedback on our manuscript, and all the suggestions offered have been immensely helpful.

We have included the reviewer comments immediately after this letter and they were answered point-by-point (blue color). All the remarks have been considered and the new version of the manuscript has been included. The corresponding changes can be visualized on “Tracked Changes” and I further declare that the revisions have been approved by all authors.

We hope the revised manuscript can be published in Metabolites, and we thank you for your continued interest in our research.

Reviewer #2:

This study has certain significance in bioactive phytochemicals in food services... However, revisions are necessary for the current version of the manuscript. The following questions to be addressed/considered may be helpful to improve the manuscript.

Major comments

Comment 1. Full detail of the author’s affiliation is needed, the institute, city, postal code, and the country.

Response: Thank you for the observation. As all laboratories mentioned in the affiliation are in the same building, we include the postal code, city and country only in the first one (affiliation a) and we repeat the others.

Comment 2. Insufficient Abstract: In the abstract, the main aim and background of the manuscript are missing, the current version it only highlights the result. In addition, it would be even better to have a sentence as a future perspective.

Response: We appreciate the comment, but we do not agree that the abstract is insufficient. Due to the word limit imposed on the abstract, unfortunately it is not possible to explore everything that our paper offers. However, we believe that the main objective of the work was mentioned through the phrase "untargeted metabolomics approach was applied in the study to evaluate the phenolic compounds (PC) by metabolomic screening in five FVB after optimizing the extraction.". The main results were briefly presented and the future perspective is described in the last sentence of the abstract, where our research group believes that this work helps in the valorization of these fruit and vegetable by-products and that the applied techniques help in the phenolic understanding of these extracts.

Comment 3. The unit/abbreviation is not mentioned before, consider defining the abbreviation when mentioned for the first time…. Please check throughout the manuscript to define the abbreviations.

Response: Thank you for the comment. All abbreviations were revised throughout the manuscript.

Comment 4. Grammar and punctuation issuers need to be addressed. I have selected/mentioned some as examples.

Response: Thank you for the comment. The manuscript has been modified and we believe these grammar/punctuation issues have been eliminated in the revised file.

Comment 5. I have a major concern about the results and discussion section. The authors describe the results and compare the results with previous studies, however, insight mechanisms are still insufficient.

Response: We, the authors, are not sure if we correctly interpreted the reviewer's suggestion. We think that by 'mechanism' the reviewer is referring to the ways in which the applied extraction methods acted on the different classes of phenolic compounds extracted at each step. If this interpretation is correct, then we would like to point out that there are passages throughout the text where these 'insights' were presented, but as we believe they have not been made clear, we have made several changes throughout the discussion section (on tracked changes). In addition, we would like to highlight some points where we explain the application of solvents and auxiliary methods:

Line 60-64: still in the introduction, we emphasize the importance of testing different solvents and in different proportions for each FVB. Subsequently, the results were presented and the discussion was based on the comparison with the literature (to assess the efficiency of the extraction), with hypotheses of possible phenolic classes extracted from each solvent and with the selection of the best extractor for each FVB.

Line 298-302: in this case, we explain how sonification as an auxiliary method can intensify this extraction, but with time control so that there is no degradation of the phenolic compounds.

Line 363-368: here we explain the effect of acid and alkaline hydrolysis to obtain these compounds, since a large part of them is bound to the cellular matrix.

All these techniques were explained, discussed and hypotheses were created. In case it was not clear in the text, we are willing to improve understanding.

Specific comments:

Abstract

Comment 1. If the unit/abbreviation is not mentioned before, consider defining the abbreviation when mentioned for the first time.

Response: We thank you for your note and the change has been made to the revised manuscript.

Comment 2. Line 20: replace ‘’ultrasound’’ with ultrasonication.’’

Response: We appreciate the suggestion and consider it very relevant to our article. We have changed this nomenclature throughout the manuscript and in supplemental material.

Comment 3. And with ‘’class’’ do you mean ‘’compounds’’, as I see in the text you’ve discussed just flavonoids, not other classes of flavonoids??

Response: In this section we mention the classes of phenolic acids and flavonoids. In our tentative identification, we worked with five different classes of phenolic compounds, namely: phenolic acids, flavonoids, other polyphenols, lignans and stilbenes (as shown in figure 3C). We observed that, for these fruit and vegetable by-products and under the conditions in which the extractions were carried out (with the solvents and equipment used), the major classes are phenolic acids and flavonoids. Still, between the extraction conditions (SM, US and TPC) there is variation between these two classes. Therefore, only these two classes were cited in the abstract.

Comment 4. Line 21-23: A complicated sentence, please revise.

Response: Thank you for the comment. We made a small modification to the sentence by changing the word “while” to “and”. In this sentence, we want to explain that the parts of the fruit/vegetable from which the by-product was obtained are important for its phenolic profile. An example is the peels samples (of carrots and chayote), which, regardless of whether they come from a different fruit/vegetable, have a similar profile. This is confirmed by the papaya seed, since this by-product is the only one among the five that comes from this part of the fruit (seed) and, therefore, has a different profile from the others. We hope that, even with this small modification, this sentence was clear in the text.

Introduction

Comment 5. Line 40: A reference is needed here, for example, you can use: https://doi.org/10.3945/an.112.002154. Gupta, V.K., Tuohy, M.G., O'Donovan, A. and Lohani, M. eds., 2015. Biotechnology of bioactive compounds: sources and applications. John Wiley & Sons.

Response: Thank you for the suggestion. We had already used a reference for this sentence (Granato, D.; et al. Functional Foods: Product Development, Technological Trends, Efficacy Testing, and Safety. Annual Review of Food Science and Technology, Vol 11 2020, 11, 93-118, doi:10.1146/annurev-food-032519-051708), but we have included the citation suggested by the reviewer.

Comment 6. Line 58-63: A complicated sentence, please revise and check the grammar.

Response: We made a small modification to the sentence, simplifying the words used to make it more understandable. The purpose of this statement is to show that plant matrices are complex and to obtain an isolated component, such as phenolic compounds, it is necessary to optimize an extraction. Therefore, in our study we applied an extraction that our research group uses conventionally and compared it with optimized methods.

Comment 7. Line 59: A reference is needed here, for example: https://doi.org/10.1016/j.foodres.2020.109854.

Response: We appreciate the important suggestion and have included the reference in the revised manuscript.

Comment 8. Line 71: A reference is needed here, for example, you can use: https://doi.org/10.1080/22297928.2016.1152912.

Response: We appreciate the important suggestion. However, this sentence already contains a citation (Rosello-Soto, E.; Koubaa, M.; Moubarik, A.; Lopes, R.P.; Saraiva, J.A.; Boussetta, N.; Grimi, N.; Barba, F.J. Emerging opportunities for the effective valorization of wastes and by-products generated during olive oil production process: Non-conventional methods for the recovery of high-added value compounds.Trends in Food Science & Technology 2015, 45, 296-310, doi:10.1016/j.tifs .2015.07.003) and we believe that this has more to do with the sentence written in the manuscript and with the plant material used in the present study.

Comment 9. Line 75-78: These are rather long sentences, better to break them down into more sentences.

Response: Thank you for the suggestion. The sentence was break into two in the revised manuscript.

Comment 10. Line 62-66: A complicated sentence, please revise and check the grammar.

Response: Thank you for the observation, but we don't think the sentence is confusing. We believe it is clear that the influence of extraction conditions (such as temperature, time, agitation speed, among other factors) can influence the profile and content of phenolic compounds. Unfortunately, we have not found another way to describe this phrase in a clearer way than what is already presented in the manuscript.

Material and methods

Comment 11. Literature references are missing for all sub-section. It would be better to cite the references that the procedure adopted.

Response: Almost all sub-topics of the methodology have a reference, except for topic 2.1 (the collection of plant material) and the optimized extraction in topic 2.2, as it was carried out for the first time in an experimental way with the objective of selecting the best extractor and time. The other topics (extraction of free and bound compounds, analyzes of antioxidant capacity and the application of metabolomics tools.

Comment 12. Additional info is needed for the table caption, most importantly significant figures.

Response: Thank you for the comment. All table information is in the footnote, including statistically significant differences.

Comment 13. In MM section, what is the quality control (QC) data? There is no mention of the QC.

Response: Quality control (QC) was prepared by pooling equal volumes of each FVB extracts. We appreciate the suggestion to include this information and have included it in the revised manuscript.

Comment 14. What is the accuracy of the instruments, recovery, LOD, and LOQ ……. These parameters are needed to report the efficiency of any analytical system.

Response: We appreciate the reviewer's suggestion, but our analytical method is a relative quantification by ion abundance. Our relative abundance for a specific ion in the sample was calculated by dividing the number of ions with a particular m/z ratio by the total number of ions detected. With this relative quantification, we do not perform these calculations, such as LOD and LOQ, and relate more abundant compounds in a sample by their ionic intensity.

Comment 15. In general, how many times you’ve recorded the data? duplicate? Triplicate?..... what you mentioned in the text is not clear, please elaborate more on this.

Response: Thank you for the observation. The injections were performed in triplicate, and we have already included this information in the revised manuscript.

Results and discussion

Comment 16. Table 1: Consider to have harmonize the significant figures for all variables.

Response: Thank you for the comment. In this table we use the symbol "*" to indicate the significant difference (p<0.05). The other values that do not have this symbol are statistically equal. This information is in the footnote of Table 1.

Comment 17. What is the difference between Table 3 and table S1? If the data ais the same consider deleting one, you don’t have to preset the data twice, despite the table 3 is summarized.

Response: Table S1 presents all phenolic compounds tentatively identified in order of retention time in the chromatogram, while Table 3 is organized with the most abundant (i.e., most representative) compounds in each FVB extract. They are different tables and, therefore, we believe that both are important in the manuscript.

Comment 18. These sections are repeating information already presented and explain things in an unnecessarily complicated way. The quality of the manuscript would benefit from the whole section being condensed, Line 236-260, Line 293-333, Line 409-433, Line 492-526.

Response: These marked lines indicate different results from different analyzes and different comparisons:

Line 236-260: addresses the selection of the best chayote peel extractor. Each FVB presented its proportion of H20 and EtOH that presented the best result. In this case, the chayote peel showed higher contents when extracted with 100% H2O.

Lines 293-333: Once the extractor was selected for each FVB, the ultrasonication times were applied. In this section, the explanation of the best time is covered.

Line 409-433: this section deals with metabolomic analyses, where the US (ultrasonication extract) is compared with the other SM, FPC, BPC and TPC, with the aim of establishing a comparison between the extractors.

Lines 492-526: each relevant phenolic compound from each FVB is discussed in this section. Furthermore, multivariate analyzes are applied to show the differences between the FVB. Unlike lines 409-433, where the focus is on extractors, this section discusses the extractor-independent FVB.

As a result, these sentences cannot be condensed.

Comment 19. Line 197-198: even though you have defined the abbreviation, nut you still repeated the full name, consider using the abbreviation you define. Please check throughout the manuscript.

Response: Thank you for the observation. These abbreviations were checked throughout the revised manuscript.

Comment 20. Figures 1, and 2. how the comparing made between the treatments and assigning the letter for the statistical difference is confusing. For example, Figure 1 a: How you can have d, ab, a, b, c, c, …..? Please elaborate more!

Response: Information about the letters assigned after statistical treatment are described in the footnotes of the figures. Both figures (1 and 2) have different letters that indicate a significant difference between extractors (Tukey, p<0.05). Example of Figure 1 - AC on carrot peel: the 50% EtOH extractor showed the highest values, followed by 75% EtOH, FPC, 25% EtOH, 100% H2O and, finally, 100% EtOH. All extractors are different from each other and therefore have different letters.

Conclusion

Comment 21. Important conclusions! However, the future perspectives for the following research are highly crucial here.

Response: We appreciate the comment. We made changes in the conclusion of the revised manuscript, and we brought the application perspectives of these extracts rich in bioactive compounds.

Round 2

Author Response

We appreciate the reviewer's comments. For better writing and therefore better understanding for the reader, we have modified the entire section 2.3: determination of antioxidant capacity. We have further described all information that was not previously clear and included a final sentence explaining how the calculations were performed.

We hope that all questions regarding this section have been resolved.

Reviewer 2 Report

The revised manuscript has improved compared to the original version. The authors tried to address most of my questions!

I was more concerned about the specific comments for the introduction and MM sections. For example, in the introduction, additional references were needed. In the MM section, LOD or LOQ for the system were missing '' the author stated: they approach qualitatively, if that is so, they need to state in the MM section ''which is not''. Anyhow, these are minor comments, I recommend the manuscript be accepted it.

Author Response

We appreciate the reviewer's comment and the recommendation to accept the manuscript.

Since it was not very clear how we do our relative quantification, we included it in one sentence at the end of section 2.4. We hope that this inclusion has made the method clearer and that it will not cause confusion among readers.

Round 3

Reviewer 1 Report

Accept in present form